# Unveiling the additive-assisted oriented growth of perovskite crystallite for high performance light-emitting diodes

Lin Zhu[1,4], Hui Cao[1,4], Chen Xue [2,4], Hao Zhang[1], Minchao Qin [3], Jie Wang[1], Kaichuan Wen[1], Zewu Fu[1], Tao Jiang[1], Lei Xu[1], Ya Zhang[1], Yu Cao [1,2], Cailing Tu[1], Ju Zhang[1], Dawei Liu[1], Guangbin Zhang[1], Decheng Kong[1], Ning Fan[1], Gongqiang Li [1], Chang Yi[1], Qiming Peng [1], Jin Chang[1], Xinhui Lu [3], Nana Wang [1✉], Wei Huang [1,2✉] & Jianpu Wang [1✉]

Solution-processed metal halide perovskites have been recognized as one of the most promising semiconductors, with applications in light-emitting diodes (LEDs), solar cells and lasers. Various additives have been widely used in perovskite precursor solutions, aiming to improve the formed perovskite film quality through passivating defects and controlling the crystallinity. The additive's role of defect passivation has been intensively investigated, while a deep understanding of how additives influence the crystallization process of perovskites is lacking. Here, we reveal a general additive-assisted crystal formation pathway for FAPbI$_3$ perovskite with vertical orientation, by tracking the chemical interaction in the precursor solution and crystallographic evolution during the film formation process. The resulting understanding motivates us to use a new additive with multi-functional groups, 2-(2-(2-Aminoethoxy)ethoxy)acetic acid, which can facilitate the orientated growth of perovskite and passivate defects, leading to perovskite layer with high crystallinity and low defect density and thereby record-high performance NIR perovskite LEDs (~800 nm emission peak, a peak external quantum efficiency of 22.2% with enhanced stability).

[1] Key Laboratory of Flexible Electronics (KLOFE) & Institute of Advanced Materials (IAM), Nanjing Tech University (NanjingTech), Nanjing, China. [2] Shaanxi Institute of Flexible Electronics (SIFE), Northwestern Polytechnical University (NPU), Xi'an, China. [3] Department of Physics, The Chinese University of Hong Kong, Shatin, Hong Kong. [4] These authors contributed equally: Lin Zhu, Hui Cao, Chen Xue. ✉email: iamnnwang@njtech.edu.cn; iamwhuang@nwpu.edu.cn; iamjpwang@njtech.edu.cn

Solution-processed perovskite light-emitting diodes (LEDs) have achieved significant progress in the past few years, boosting the external quantum efficiency (EQE) to >20%[1–5]. Defect passivation has been proven to be an effective strategy to enhance the optoelectronic property of perovskites, which can be easily achieved by introducing additives in precursor solution[1,5,6]. Various additives, including organic molecules[7,8], metal halides[9–11], and ammonium halides[12–14], have been widely used to passivate defects in perovskite solar cells and LEDs. Apart from the defect passivation effect, additives play a key role in the crystallization process of perovskite[7,15,16], which is also important for the resultant device performance. However, a deep understanding of how the additives influence the crystallization process of perovskites is lacking, which prevents further design of additive for high-quality perovskites. Here we demonstrate a general additive-assisted crystal formation pathway for FAPbI₃ perovskite with vertical crystallographic orientation. We found that the crystallization process of perovskite is determined by the interaction of the additives with formamidinium iodide (FAI) in the precursor solution. Additives with amino group (–NH₂) can form an intermediate structure with FAI and deplete free FAI in the precursor, which then facilitate the formation of vertical growth of low-dimensional perovskite sheets during the early stage of spin-coating process, leading to vertically oriented FAPbI₃ perovskites with high crystallinity and less grain boundaries during annealing process. In contrast, additives possessing weak interaction with FAI, such as carboxyl group (–COOH) molecules, have minor effect on the crystallization process, which mainly passivate the defects of halide vacancy. Based on this understanding, we further demonstrate a champion perovskite LED with a record peak EQE of 22.2% with much improved stability by incorporating a new additive with multifunctional groups, 2-(2-(2-aminoethoxy)ethoxy)acetic acid (AEAA), to both control the crystallinity and passivate various defect sites.

## Results

**Perovskite films based on various additives**. We fabricated three-dimensional (3D) FAPbI₃ perovskites by introducing additives with various functional groups to the precursor solutions, i.e., pentan-1-amine (PAM) and pentanoic acid (PAC). The perovskite films are prepared by spin-coating precursors of additive:FAI:PbI₂ with a molar ratio of $x$:2.4:1, followed by annealing at 100 °C for 20 min. We note that, in order to boost the radiative recombination in perovskite LEDs, usually a low concentration of precursor solution is used to obtain very thin perovskite film[1,5,17,18]. The scanning electron microscope (SEM) measurement shows that the perovskite layer without additive is discrete and irregular-shape grains with size of between 50 and 500 nm. And X-ray diffraction (XRD) measurement shows very weak diffraction peaks at around 14° and 28° (Fig. 1). Interestingly, the PAM-based perovskite layer exhibits faceted shape of crystallites. As the PAM ratio increased, the perovskite grains become more regular and larger, and the diffraction peaks corresponding to the α-phase FAPbI₃ become stronger and narrower (Supplementary Fig. 1), indicating larger crystallites and preferentially perpendicular orientation with respect to the substrate. This result indicates that the PAM-based perovskite has the potential to achieve low trap density film[19], which can lead to high-performance device. Importantly, the perovskite layers prepared with other amino group molecules also show enhanced crystallinity, regardless of aryl or alkyl (Supplementary Fig. 2). This suggests that it is a universal effect for amine additives in FAPbI₃, although there are minor variations in film morphology and crystallinity. In contrast, PAC has almost no impact on the crystallization of FAPbI₃ perovskite layer, exhibiting similar SEM

images and crystallinity as perovskites without additive. Varying the chain length and amount of additive with carboxyl group will also not cause significant change in morphology and crystallinity of perovskite films (Supplementary Figs. 2 and 3). We further investigated additives with other end groups, i.e., pentanenitrile, pentane-1-thiol, and pentan-1-ol, which also result in similarly unoriented perovskite layers as without additive (Supplementary Fig. 2).

**The roles of additive in the crystallization process of perovskites**. In order to understand how the additives with different end groups affect the crystallization kinetics of perovskites, we performed synchrotron-based in situ grazing-incidence wide-angle X-ray scattering (GIWAXS) measurements during the spin-coating process. The two-dimensional (2D) GIWAXS patterns were measured at 1 s per frame. As shown in Fig. 2a, the PAM-based perovskite shows a characteristic α-phase scattering ring at $|q| = 1.00$ Å$^{-1}$, which arises from the [PbI₆]⁴⁻ octahedron in corner–corner shared connection[20,21]. The ring is concentrated in both the out-of-plane ($q_z$) and in-plane ($q_r$) directions with much higher intensity in the $q_z$ direction, suggesting a preferential stacking of the vertical oriented [PbI₆]⁴⁻ octahedrons along the surface normal direction. In contrast, the corresponding scattering ring of the PAC-based perovskite appears isotropic in polar angle (Fig. 2b), implying a random crystalline orientation. The in situ GIWAXS intensity profiles versus frame numbers along the $q_z$ and $q_r$ directions are plotted in Fig. 2c–f and Supplementary Fig. 4. The 2D GIWAXS patterns measured at different time points are summarized in Supplementary Fig. 5. The PAM-based perovskite forms δ-phase FAPbI₃ initially at $|q| = 0.80$ and $1.80$ Å$^{-1}$ (around 8 s) with the [PbI₆]⁴⁻ octahedron in face–face shared connect mode[20,22]. It quickly transforms to the corner–corner shared [PbI₆]⁴⁻ octahedrons, exhibiting the α-phase peak located at $|q| = 1.00$ Å$^{-1}$. Remarkably, it is observed that the peak appears slightly earlier in the $q_z$ direction (9 s) than in the $q_r$ direction (11 s), implying that the perovskite crystals build up in the out-of-plane direction first. This delicate anisotropic launch of crystal growth could lead to the cumulative outcome of highly oriented PAM-based perovskite thin layers. For the PAC-based perovskite, the α-phase peak appears simultaneously (7 s) in the $q_z$ and $q_r$ directions, signifying an isotropic crystallization. It is consistent with the observed uniform scattering ring in the resultant film. Remarkably, the in situ GIWAXS measurement suggests that the crystallinity of the perovskite layers is mainly determined at the very early stage of the spin-coating process. Therefore, we can infer that the difference of crystallization pathways of perovskites originates from the different chemical interaction in precursor solutions when different additives are used.

We investigate the different chemical interactions in the PAM– and PAC–perovskite precursor solutions by performing ¹H nuclear magnetic resonance (¹H NMR) measurements. Figure 3 shows that the resonance signal attributed to the methylene group of PAM (α-CH₂ protons, indicated with *) appears at $\delta = 2.46$ ppm, which shifts to low field by $\Delta\delta = 0.51, 0.87, 0.86$ ppm in the PAM•PbI₂, PAM•FAI, and PAM•PbI₂•FAI deuterated dimethyl-formamide (DMF) solutions, respectively. This indicates that the hydrogen bond between PAM and FAI is stronger than the coordination bond between PAM and PbI₂ in perovskite precursor solution, which was also confirmed by the titration experiment of FAI in PAM•PbI₂ solution (Supplementary Fig. 6a). Moreover, after adding PAM into the FAI solution, the N-H protons of FAI ($\delta = 9.25$ ppm) become broader and move to high field, with trace-free FAI signal, indicating that most FAI forms an intermediate with PAM. The titration experiments of PbI₂ in

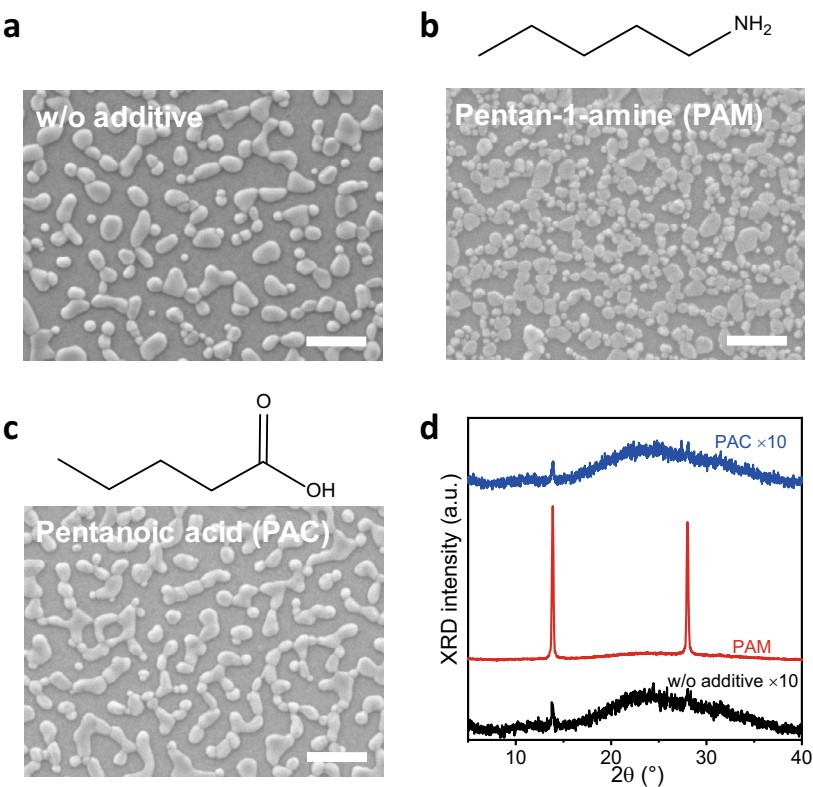

**Fig. 1 Morphology and crystallinity of FAPbI₃ perovskites with various additives. a–c** SEM images of FAPbI₃ perovskites without (**a**) or with pentan-1-amine (**b**) and pentanoic acid (**c**) as additives (scale bar: 1 μm). The chemical structures of additives are shown on top of the SEM images. The ratios of additives are PAM:FAI:PbI₂ = 0.5:2.4:1 and PAC:FAI:PbI₂ = 0.9:2.4:1, respectively. **d** XRD patterns. It shows typical 3D α-phase PAPbI₃ and no peaks for 2D phase.

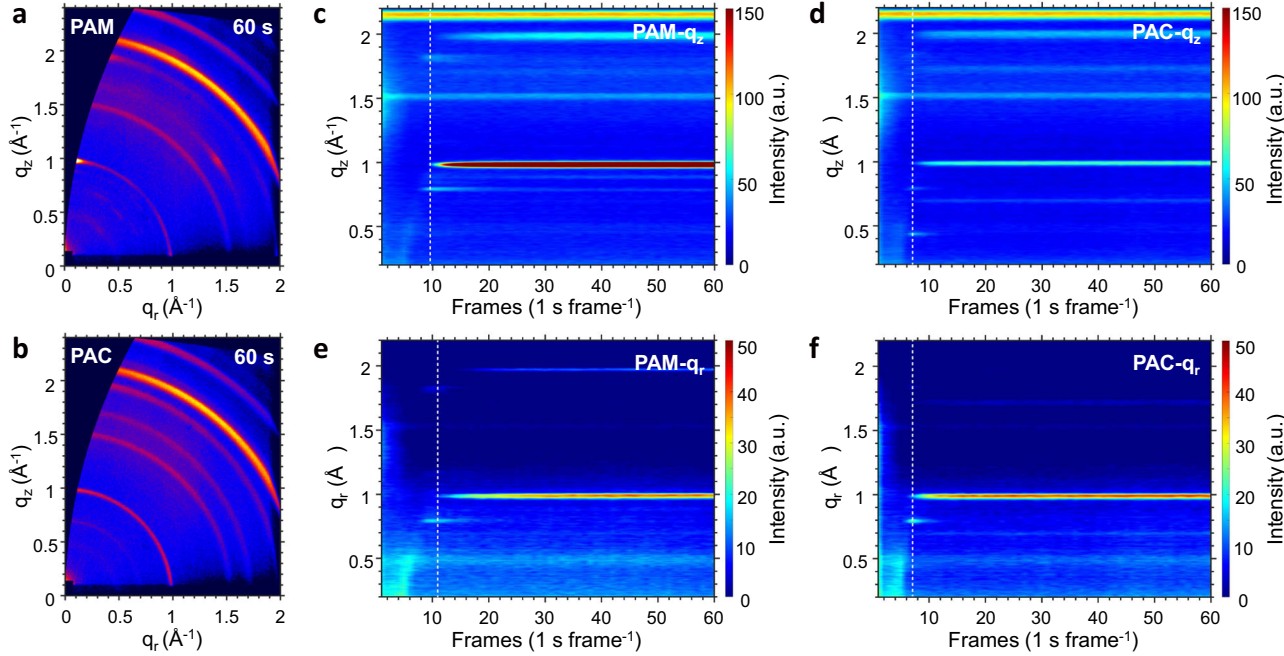

**Fig. 2 Time-resolved GIWAXS profiles of FAPbI₃ perovskites with PAM and PAC. a, b** 2D GIWAXS patterns of PAM-based (**a**) and PAC-based (**b**) perovskites recorded after 60 s of the spin-coating process. **c, d** The false-color intensity maps versus $q_z$ and frames (1 s frame⁻¹) for perovskites with PAM (**c**) and PAC (**d**). **e, f** The false-color intensity maps versus $q_r$ and frames (1 s frame⁻¹) for perovskites with PAM (**e**) and PAC (**f**).

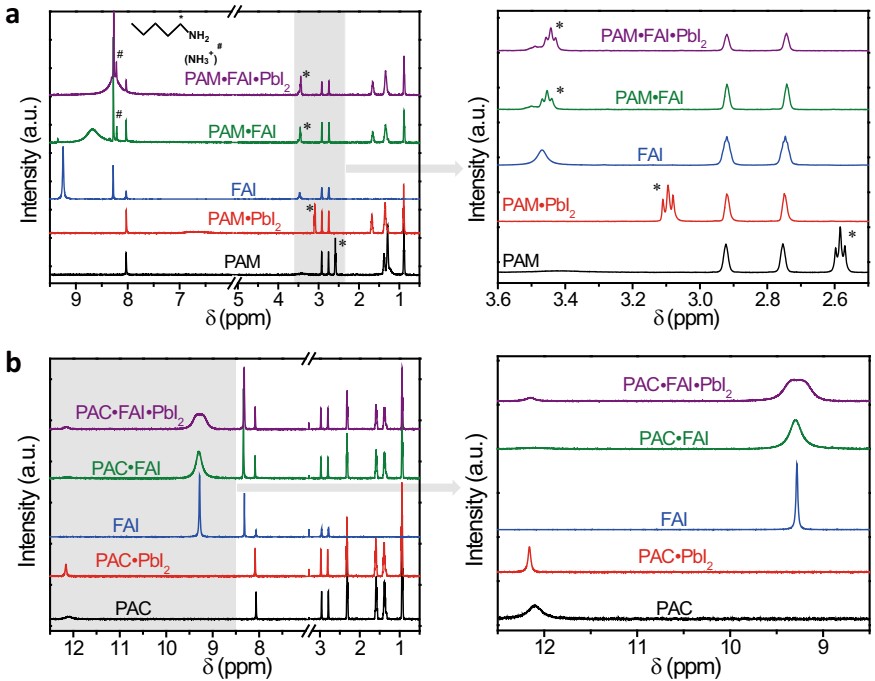

**Fig. 3 ¹H NMR spectra (DMF-$d_7$, 500 MHz) of perovskite precursor solutions. a** PAM-based perovskite solution. **b** PAC-based perovskite solution.

PAM•FAI solution also show the broadened signal of N-H protons of FAI (Supplementary Fig. 6b), suggesting that iodide ions take part in the intermediate complex {PAM + FA + [I + FA]$_n$}$^+$. The resulted complex was further identified by the electrospray ionization–time-of-flight mass spectrometry (ESI-TOF MS), which shows a peak at $m/z = 115.12254$ corresponding to [PAM + FA-NH$_3$]$^+$ for the intermediate {PAM + FA + [I + FA]$_n$}$^+$ (Supplementary Fig. 6c). Besides, the appearance of $\delta = 8.20$ ppm assigned to the NH$_3^+$ (indicated with #) in PAM•FAI and PAM•PbI$_2$•FAI solutions shows that FAI can change PAM to PAM$^+$ in the precursor solution. In contrast, PAC has little effect on the ¹H NMR spectrum of the perovskite precursor solution, except for the broadened signal of N-H proton of FAI (Fig. 3b). This indicates the hydrogen bond between FAI and PAC is much weaker than with PAM. Meanwhile, the ESI-TOF MS spectra show that no new complex was generated in the PAC-based perovskite solution (Supplementary Fig. 6c). Therefore, we can conclude that the strong hydrogen bond between PAM and FAI results in the formation of an intermediate complex and the deficit of free FAI in the precursor solution, which is not observed in the PAC-based perovskite solution. Moreover, the NMR spectra also confirm that various –NH$_2$ group additives with different sizes and structures have similar strong interaction with FAI solution, while the –SH, –OH, –CN group additives result in negligible variations of FAI signals similar as PAC (Supplementary Fig. 6d, e). These results suggest that the chemical interaction between the functional group of additive and FAI plays a critical role in the crystallization process of perovskites, rather than the size and structure of additive molecule.

Based on the above accumulated observations, we propose that the distinct crystallization in PAM-based perovskite originates from the strong hydrogen bond network {PAM + FA + [I + FA]$_n$}$^+$. At the beginning of spin-coating process (Fig. 4a, Stage I), the low polarity of [PAM + FA]$^+$ will stand ordering along the interface between the perovskite precursor and air with the alkenyl head directing to the air, which then act as scaffolds for the initially vertical growth of perovskite. Owing to the deficit of

free FA$^+$ in the precursor solution, the homogeneous nucleation within the supersaturated perovskite solution, which could lead to randomly oriented perovskite crystals, is suppressed[23,24]. There is continuously vertical growth of PAM-based perovskite crystal, resulting in an intermediate complex containing vertically oriented 2D perovskite layers with PAM$^+$ and [PAM + FA]$^+$ as large organic cations (Stage II). This also can be verified by the Bragg spots at $|q| \approx 0.45$, 0.7, and 1.7 Å$^{-1}$ from the in situ GIWAXS measurement (Supplementary Fig. 5). Then during annealing process, there is in-plane growth with controlled orientation as the releasing of free FA$^+$ (Stage III). In the meantime, the residually protonated PAM$^+$ in the perovskite layer can passivate the FA vacancy defects[12,25]. In the PAC-based perovskite, due to the weak interaction between PAC and FA, there are plenty of free FA$^+$ in the colloidal state, which can induce homogeneous nucleation[23], leading to earlier crystal growth and randomly oriented perovskites (Fig. 4b). Consequently, the –COOH group has little effect on the crystallization process but mainly plays the role of passivating the iodine-vacancy defects of FAPbI$_3$ through coordinating with the unsaturation lead (Supplementary Fig. 7).

The above proposed perovskite formation pathway can be further confirmed by the analysis of perovskite crystallites during annealing process (Supplementary Figs. 8 and 9). As annealing time increases, both the PAM- and PAC-based perovskites show a tendency to form larger grains and the intermediately low-dimensional crystalline phases are gradually transformed into 3D perovskites, exhibiting typically crystalline and optical features of 3D FAPbI$_3$. However, XRD measurement shows that the PAM-based FAPbI$_3$ perovskite shows enhanced crystallinity with the annealing, while that of PAC-based perovskites exhibits negligible change. This result is consistent with the proposed growth mechanism in Fig. 4, which shows that the α-phase FAPbI$_3$ forms due to the release of large amount of free FA$^+$ with the PAM-based FAPbI$_3$ perovskite layer during annealing, while the crystallographic orientation is already determined before the annealing process. Moreover, we point out that the additives will remain on top of the substrates and locate in between the

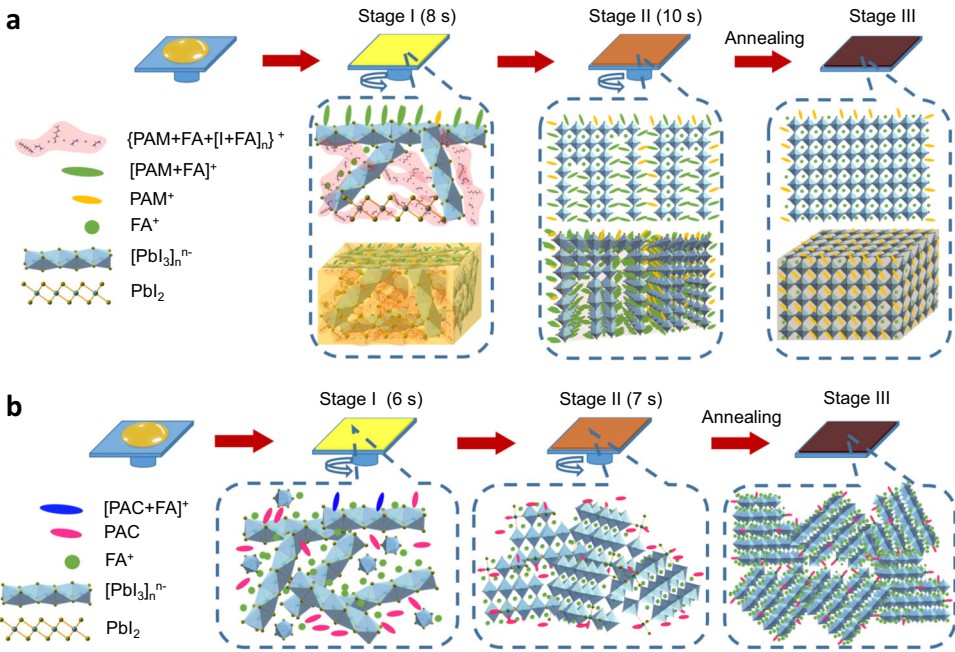

**Fig. 4 Schematic illustration of growth pathways of FAPbI₃ perovskites with additives. a** PAM-assisted oriented growth. **b** Isotropic growth with PAC additive.

perovskite domains, which can prevent the LED device from short circuit, as confirmed by the early study[1,18].

We then study the optoelectronic properties of FAPbI₃ perovskites with PAM and PAC (Supplementary Figs. 1 and 3). As investigated by the transient photoluminescence (PL) and PL quantum efficiency (PLQE) measurement, both PAM- and PAC-based perovskite show reduced trap densities and enhanced optical properties, indicating that the defect passivation is effective for both PAM and PAC additives. We then fabricated devices with the structure of indium tin oxide (ITO)/polyethylenimine ethoxylated (PEIE)-modified zinc oxide (ZnO; 30 nm)/ perovskite (~50 nm)/poly(9,9-dioctyl-fluorene-co-N-(4-butylphenyl)diphenylamine) (TFB; 40 nm)/molybdenum oxide (MoO$_x$; 7 nm)/gold (Au; 60 nm). The average peak EQEs for PAM- and PAC-based perovskite LEDs are $14.2 \pm 0.9$ and $12.5 \pm 1.2\%$, respectively. Although there is no significant difference in device efficiency between them, we found that the device lifetime can be significantly improved with the PAM-based oriented perovskites. With simple glass-epoxy encapsulation, the PAM and PAC devices show half-lifetime ($T_{50}$) of 9.5 and 5.4 h, respectively (Supplementary Fig. 10). We believe that the better stability of PAM-based device is due to its high crystallinity with less grain boundary, since the degradation of the perovskites are usually starting from the grain boundary[26].

**High-performance LEDs based on perovskite with multifunctional group additive.** In order to achieve high-performance perovskite LEDs, it requires perovskites with low defects and high PLQE, as well as good crystallinity to improve the stability. Considering the complementary functions of –NH₂ and –COOH groups, we further introduce a new additive, AEAA, which has both the two groups. The 0.5-ratio AEAA perovskite layer shows a high peak PLQE of 78% compared to other 3D perovskites[1,5] and the PLQE is even higher than 60% at an extremely low excitation intensity of 0.01 mW cm⁻² (Supplementary Fig. 11a), indicating a very low defect density due to the amino-group-induced high crystallinity and effective passivation of various

defect sites. We can obtain a trap density of $(9.1 \pm 5.7) \times 10^{12}$ cm⁻³ by fitting the transient PL data (Supplementary Fig. 12)[27], which is close to the trap density of thin single crystal perovskites[19] and one order of magnitude lower than the $(2.0 \pm 0.7) \times 10^{14}$ and $(2.4 \pm 1.7) \times 10^{14}$ cm⁻³ of PAM- and PAC-based perovskites, respectively.

We believe that the low trap density and high PLQE of the AEAA perovskite is due to both the effects of oriented growth of perovskite crystallite and better passivation of various defects. The ¹H NMR spectra show that there is strong hydrogen bond between AEAA and FAI in the perovskite precursor solution through the amino group (Supplementary Fig. 11c). This strong chemical interaction can result in an intermediate complex $\{AEAA + FA + [I + FA]_n\}^+$, which facilitates the vertical growth of perovskite from the liquid/air interface similar as other amino group molecules. The oriented growth of the AEAA perovskite can be confirmed by SEM and XRD measurements (Supplementary Fig. 11d, e). Supplementary Fig. 13 shows that, during thermal annealing, the AEAA-based film is gradually transformed into 3D perovskites from intermediately low-dimensional phases. The AEAA-based FAPbI₃ layer exhibits faceted platelet with large and highly oriented perovskite crystallites, which is beneficial to a low density of defects due to the same orientation and less grain boundaries[19,28]. In addition, the X-ray photoelectron spectroscopic (XPS) spectra of AEAA-based perovskite film show that the Pb 4f peak shifts to the lower binding energy (BE), along with the shift of C=O peak to higher BE with increased content (Supplementary Fig. 7), which suggests the coordination between carbonyl group and unsaturation Pb. Moreover, the intensity of peak at 533.0 eV increases, which may originate from the interaction of C-O-C species with FA⁺ or unsaturation Pb. Importantly, the ¹³C NMR measurements of AEAA-based perovskite precursor solution further confirm that the C-O-C and COOH groups of AEAA can interact with both PbI₂ and FAI (Supplementary Fig. 11f), which suggests that the carboxyl group can better passivate iodine-vacancy defects with the assistance of ether group through the formation of chelate ring (Supplementary Fig. 11g)[29,30]. Therefore, the above results unambiguously

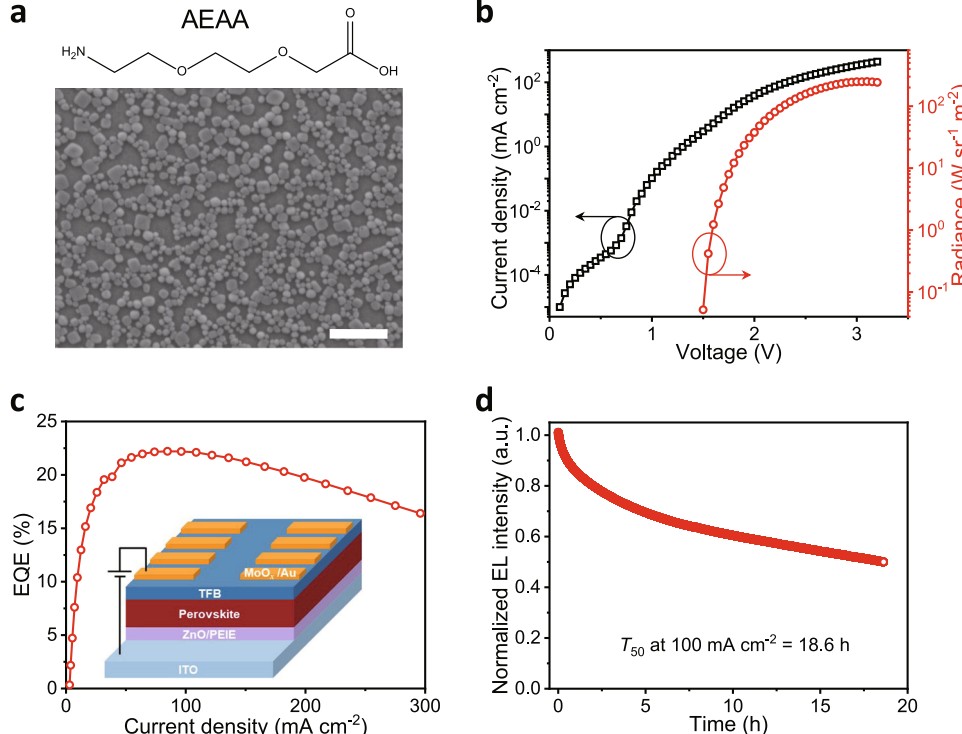

**Fig. 5 Characteristics of AEAA-based perovskite layers and devices. a** SEM image (scale bar: 1 μm). The chemical structure of AEAA is shown on top of the SEM image. **b** Current density and radiance versus voltage. The device shows small leakage current due to the formation of thin organic layer between discrete crystallites[1]. **c** Dependence of EQE on current density. **d** Stability of devices measured at a constant current density of 100 mA cm$^{-2}$.

suggest that the AEAA perovskite can combine the advantages of both the effects of oriented growth of perovskite crystallite and better passivation of various defects, which can be illustrated in Supplementary Fig. 11h.

The champion 0.5-ratio AEAA-based device yields a brightness of 250 W sr$^{-1}$ m$^{-2}$ at a low voltage of 3.1 V (Fig. 5 and Supplementary Fig. 14) and exhibits a peak EQE of 22.2% at a current density of 85 mA cm$^{-2}$ and a radiance of 93 W sr$^{-1}$ m$^{-2}$, representing a record efficiency for perovskite LEDs (Supplementary Table 1). In addition, the discrete crystallites of our perovskites are beneficial for light outcoupling efficiency of electroluminescent (EL) devices, which results in an outcoupling efficiency of 29.1 ± 2.4% by using the calculation method developed previously[1]. Based on the outcoupling efficiency, we can estimate that the peak EQE corresponds to an internal quantum efficiency of about 76%, which is consistent with the PLQE. The EL device also has high reproducibility, showing an average EQE of 19.4 ± 1.0% (Supplementary Fig. 14d). We believe the device efficiency can be further boosted by optimizing additives to facilitate the orientated growth, passivate defects, and tune the microstructure of perovskite layer. More importantly, the improved quality and reduced grain boundaries of perovskite layers contributes to better device stability, leading to a $T_{50}$ of 18.6 h at a very high constant current density of 100 mA cm$^{-2}$. The EL emission peak locates at ~800 nm, which remains constant at various bias voltages and during lifetime test (Supplementary Fig. 14). So we believe that the perovskite phase remains stable during the lifetime measurement.

## Discussion
By tracking the chemical interaction in the precursor solution, crystallographic evolution during spin-coating and annealing process of perovskites, we revealed a universal additive-assisted crystal formation pathway for FAPbI$_3$ perovskite with vertical crystallographic orientation. Based on this understanding, we then demonstrate record high-performance perovskite LEDs by

using AEAA additive, which can distinctly facilitate the orientated growth of perovskite and effectively passivate defects, leading to perovskite layer with high crystallinity and low defect density. Our findings suggest a universally selective rule of additive for high-quality light-emitting perovskites.

## Methods
**Synthesis and material preparation**. The additive-based FAPbI$_3$ precursor solutions were prepared by dissolving additives FAI and PbI$_2$ with a molar ratio of $x$:2.4:1 in DMF (7 wt.%). The colloidal ZnO nanocrystals were synthesized according to previously reported procedures with modifications[31]. PEIE (Sigma-Aldrich, 37 wt.% in H$_2$O), TFB (American Dye Source, molecular weight >30,000), and MoO$_x$ (Alfa Aesar, 99.95%) were used as received.

**Device fabrication**. Devices were fabricated on ITO-coated glass substrates. ZnO nanocrystals were spin-coated at 4000 rpm for 45 s and annealed at 150 °C. PEIE was spun-cast from a 0.4 wt.% solution in 2-methoxyethanol at 5000 rpm. Next, solutions of perovskite precursor were spin-coated onto the PEIE-treated ZnO films at a speed of 3000 rpm for 45 s, followed by annealing at 100 °C. The TFB layers were prepared by spin coating the 8 mg mL$^{-1}$ solution in m-xylene at 2000 rpm. Finally, the MoO$_x$ and Au layers were thermally evaporated through a shadow mask, which can define the device area of 3 mm$^2$ by the overlapping of ITO and Au electrodes.

**Device characterizations**. The perovskite LEDs were measured in a nitrogen-filled glovebox by using a Keithley 2400 source meter and a fiber integration sphere (FOIS-1) couple with a QE65 Pro spectrometer. A detailed description can be found elsewhere and the measurement system was cross-checked at the University of Cambridge (Richard H. Friend group) and Zhejiang University (Yizheng Jin group)[1,32]. The devices were measured from zero bias to forward bias at a rate of 0.05 V s$^{-1}$ without pre-bias. The stability of devices with simple glass-epoxy encapsulation was measured in air.

**Film characterizations**. The morphology of perovskite layers was collected by SEM (JEOL5 JSM-7800F). The XRD data were obtained by using a Bruker D8 Advance. The absorbance and PL spectra were measured by using a spectro-photometer with an integrating sphere (PerkinElmer, Lambda 950) and a QE65 Pro spectrometer (a 445 nm continuous wave (CW) laser as an excitation source), respectively. The PL spectra were measured by using a QE65 Pro spectrometer and

a 445 nm CW laser as an excitation source. The PLQE of perovskite layers was measured according to previously reported method by combining CW laser, optical fiber, spectrometer, and integrating sphere[33]. The time-resolved PL measurements were performed by using an Edinburgh Instruments spectrometer (FLS980) and a 633 nm pulsed laser as an excitation source. The $^1$H NMR and $^{13}$C NMR spectra were recorded on a Bruker AVANCE AV-500 spectrometer. Mass spectra were recorded on an Agilent 6230 Series Accurate Mass TOF using ESI. GIWAXS measurements were conducted at 23A small- and wide-angle X-ray scattering beamline at the National Synchrotron Radiation Research Center, Hsinchu[34]. A C9728DK area detector was used to collect the scattering signals and the wavelength of X-ray was 1.240 Å (10 keV). The sample-to-detector distance (186.8 mm) was calibrated with a lanthanum hexaboride (LaB$_6$) sample. The incident angle was kept at 1° to enhance the signal resolution with a frame exposure time of 1 s. After the perovskite precursor was dropped on the substrate, concomitant GIWAXS measurements and sample spinning could be triggered simultaneously, and the spin-coating process was conducted in an air-tight chamber under N$_2$ flow. There was no visible evidence of X-ray damage on the sample after measurements. XPS measurements were carried out by using an ESCA PHI-5000 VersaProbe (ULVAC-PHI Inc., Japan) with Al Kα anode ($E = 1486$ eV). All films were prepared on Si/ZnO/PEIE substrates except the AEAA sample. The pressure in the analysis chamber was maintained at or below $3.0 \times 10^{-9}$ Torr during the measurements. All BEs were referenced to the saturated C 1$s$ peak at 284.8 eV.

**Simulations**. We performed 3D finite-difference-time-domain simulations to calculate the outcoupling efficiency[1]. The 2D fast Fourier transform shows that 0.5-ratio PAM, 0.9-ratio PAC, and 0.5-ratio AEAA films have periods of 312–1213, 328–1021, and 269–867 nm, respectively, which correspond to outcoupling efficiencies of 27.5±3.0, 28.6±2.6, and 29.1±2.4% in LED devices.

## Data availability

The data that support the finding of this study are available from the corresponding author upon reasonable request.

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

## Acknowledgements

This work is financially supported by the Major Research Plan of the National Natural Science Foundation of China (91733302), the Natural Science Foundation of Jiangsu Province, China (BK20180085), the National Natural Science Foundation of China (61875084, 61922041, 61961160733, 51703094, 61905109, 11804156, 51972171, 61935017), the National Science Fund for Distinguished Young Scholars (61725502), the Synergetic Innovation Center for Organic Electronics and Information Displays, and the beam time and technical support provided by 23A SWAXS beamline at NSRRC, Hsinchu. M.Q. and X.L. acknowledge the financial support from Research Grant Council of Hong Kong (RGC) (14314216).

## Author contributions

Jianpu Wang had the idea for and designed the experiments. N.W., Jianpu Wang, and W.H. supervised the work. H.C. carried out device fabrication and characterizations with the assistance of Jie Wang, D.L., G.Z., and Y.C. Z.F. conducted the optical measurements with the assistance of Q.P., Y.Z., K.W., L.X., D.K., and N.F. L.Z. conducted the NMR and mass spectra measurements with the assistance of J.Z., Jie Wang, G.L., and C.Y. C.X., T.J., and C.T. performed the XRD measurements and analyzed the data. H.Z. carried out SEM measurements. C.X. analyzed the XPS data with the assistance of J.C. M.Q. carried out the GIWAXS measurements and X.L. supervised this characterization. N.W. wrote the first draft of the manuscript and Jianpu Wang provided major revisions.

## Competing interests

The authors declare no competing interests.
