## [Peer Review File · Nature Communications]

Unveiling the additive-assisted oriented growth of perovskite crystallite for high performance light-emitting diodesEditorial Note: This manuscript has been previously reviewed at another journal that is not operating a transparent peer review scheme. The manuscript was considered suitable for publication without further review at *Nature Communications*.

REVIEWER COMMENTS

Reviewer #1 (Remarks to the Author):

In this revised version of the manuscript, XRD and XPS analysis were well-conducted for perovskite films using AEAA, PAM, and PAC as an additive, and showed that combinational result of the amine's crystallinity improvement effect and acid's passivating effect can result in a higher quality perovskite emitter with low trap density. This work is still considered to be a combination approach of amine, acid, and previously reported ether group, but this work provided the working mechanism of widely used additive molecules for highly efficient PeLEDs, so it can be published in *Nature Communications*.

Reviewer #4 (Remarks to the Author):

In this manuscript, Cao et al. delivered a record-high performance near-infrared PeLED with EQE of 22.2%. The authors systematically investigated the different additive-assisted chemical interactions in the perovskite precursor solutions, and thoroughly compared the different additive-assisted crystallization mechanism for FAPbI₃ perovskites. Based on these, they applied a new additive with multi-functional groups to facilitate the oriented growth of perovskite and passivate defects, and thus reported a perovskite film with high crystallinity and low defect density. The work provides a reasonable additive-assisted crystal formation pathway for FAPbI₃ perovskites, which will be useful to researchers in related fields. Overall, the manuscript is well conceived and well-written, the characterizations can support the conclusion. Accordingly, the reviewer recommends the manuscript published in *Nature Communications* after the following issues have been addressed

1. The authors claimed that additives with multi-functional groups can facilitate the vertical orientation of perovskite and passivate defects, resulting in highly efficient PeLEDs. However, there is no clear evidence to confirm that the vertical orientation of 3D FAPbI₃ perovskite is a key factor in achieving high-efficiency PeLEDs. Therefore, the author should provide more details to explain why the vertical orientation is highly related with device performance.
2. In Line 64, the authors should attach related reference and add more explanation to demonstrate the relationship between the preferentially perpendicular orientation to substrate and the device performance.
3. In Line 171-172, considering the functions of -NH₂ and -COOH groups, the amino acid additives, such as 5-AVA (used in the authors' previous work, *Nature* 562, 249–253 (2018)), will be supposed to be beneficial for obtaining high-efficiency device. Why the authors selected AEAA? Do the additional two oxygen (O) atoms have some extra effect on the crystallization or passivation? The authors should provide more detailed measurement to demonstrate this issue.
4. Xu et al. reported that oxygen (O) atoms in the amino-functionalized additives (EDEA) would reduce the hydrogen-bonding ability (*Nat. Photonics* 13, 418–424 (2019)). The structure of AEAA is quite similar with the EDEA, and the only difference is that the one of the end groups (-NH₂) was replaced by carboxyl (-COOH). Thus, it is important to emphasize the distinctiveness of AEAA.

Reviewer #5 (Remarks to the Author):

The present manuscript unveils the perovskite crystallization process and the additive role in achieving the perpendicular nanocrystals growth in favor of developing efficient Light emitting diodes (LEDs). Amine, carboxylic acid and amine + carboxylic acid functional group additives influence on 3D perovskite crystallization and passivation were optimized and characterized to form efficient

The quality of the manuscript is good, despite it possess some research gaps and lack of scientific evidences with respect to author's claims. Major revisions needed along with considerable scientific evidences and discussions.

Followings have to be addressed,

1. Why emission of designed LEDs is limited to 800nm. Could authors employ this crystallization directing additives in achieving blue LEDs?
2. The achieved EQE of 19.4% is good, how such achieved EQE can be related with crystallization and passivation alone? Is there any other possible scientific reasons behind the EQE achievement?
3. Is vertical growth of perovskite 3D crystals merely related to the H-bond interaction alone? If so, why the authors couldn't achieve the directional growth with other amine additives?
4. EQE values are still confusing, in abstract it is stated as 19.4% whereas in introduction it is mentioned as 22.2%. Authors should finalize their EQE value and discuss with plausible explanation.
5. I suggest plotting luminance vs voltage along with radiance, so that it would be better to contrast the LED performance with other literary works.
6. It is vital to frame the comparison table for the present study with recently published literatures in terms of luminance, EQE, current efficiency, and stability to pronounce the importance.
7. Various amino group passivating agents were presented in the supporting figure, among which why do authors fix the certain agent on basis of what criteria? From XRD plots, it certainly shows the crystallization differences. Authors should compare the crystallization of the mentioned passivation agents with various doping ratios because one particular doping condition is insufficient in understanding their role in crystallization.
8. Along with the various amino passivating agent doping ratios, crystallite size monitoring with XRD peaks is demanding to reveal the crystallization effects.
9. By the way, the as-compared carboxylic acid agent effects on the morphology and crystalline features also lags with various doping ratios. The data obtained with various passivating agents remains unsatisfactory and it remains complex with no clear outputs.
10. Why authors considering only the end group effects while choosing passivating agents. What will be the effects of the other mid chain groups on perovskite crystallization?
11. The GIWAXS data recorded for PAM and PAC time varies for 7,12s and in particular, why the time variation observed with PAM and PAC agents?
12. Why particularly stronger hydrogen bond exists between PAM and FAI? If so other amine additives failed to exhibits such characters, why it occurs predominant in this case?
13. Why titration procedure for FAI equivalent is fixed to 6 and why PbI₂ equivalent is fixed to 2.2 in observing the NMR studies?
14. How the result do suggests the chemical interaction extent for different amine additives? How to scale the interaction factor by the additives? According to author's discussion, the interaction remains intact, irrespective of chain length aromatic and aliphatic nature of the additives? Kindly recheck the FTIR peaks and discuss with more scientific backgrounds for audience better understanding.
15. Does only the chemical interaction plays role in deciding the out of plane deformation. It is strange that there is no structural and geometric conformational contribution from the added components.
16. The scheme didn't work satisfactory in clearing the selective vertical growth of perovskite. It is highly recommended that addition of chemical equations in achieving the molecular ordering stage 1, 2, and 3. How do authors affirm the release of FA⁺ in stage 3. The total Figure 4 looks like a mystery as it didn't give more scientific insight with the process involved along with the characterization evidences.
17. Authors claims that Carboxylic acid group influence in crystallization is little. The interaction between carboxyl and lead is evidenced with XPS peaks. Why such interaction didn't influence the crystallization, whereas PAM interaction influenced the crystallization significantly. Moreover, I'm curious on the selective passivation on the iodine site instead of other sites present in the perovskite matrices.
18. The device stability results are compromising with glass-epoxy encapsulation. What would be the stability of LED device without any encapsulation, such figures can add value to the research carried out.
19. The light outcoupling efficiency attainment elevates the efficiency. How such outcoupling contributions were studied. The methodology wasn't provided and the characterization regarding those remains vague.

20. Lack of studies such as PL, lifetime, EL data and how the optimized color purity achieved with respect to crystallization control. Those data along with narrative flow and collective recent references can enhance the quality considerably.

21. How does the simple traditional 3D perovskite blending methodology worked efficiently, as there are numerous existing methodologies (2D, quasi 2D, nanowires, nanorods and quantum dots) works influential in achieving highly efficient and stable LEDs. How do authors will employ this strategy in establishing their future works? It is mandatory to describe the bottlenecks associated and how it can be solved in the upcoming works?

After careful consideration and reviewing process, I suggest major revisions on the above for better understanding and clarifications. All the best for the revision.

Point-by-Point Response to Referees

Reviewer #1:

Comment #1: In this revised version of the manuscript, XRD and XPS analysis were well-conducted for perovskite films using AEAA, PAM, and PAC as an additive, and showed that combinational result of the amine's crystallinity improvement effect and acid's passivating effect can result in a higher quality perovskite emitter with low trap density. This work is still considered to be a combination approach of amine, acid, and previously reported ether group, but this work provided the working mechanism of widely used additive molecules for highly efficient PeLEDs, so it can be published in Nature Communications.

Response: We thank the reviewer for the positive comment.

Reviewer #4:

Comment #1: In this manuscript, Cao et al. delivered a record-high performance near-infrared PeLED with EQE of 22.2%. The authors systematically investigated the different additive-assisted chemical interactions in the perovskite precursor solutions, and thoroughly compared the different additive-assisted crystallization mechanism for FAPbI₃ perovskites. Based on these, they applied a new additive with multi-functional groups to facilitate the oriented growth of perovskite and passivate defects, and thus reported a perovskite film with high crystallinity and low defect density. The work provides a reasonable additive-assisted crystal formation pathway for FAPbI₃ perovskites, which will be useful to researchers in related fields. Overall, the manuscript is well conceived and well-written, the characterizations can support the conclusion. Accordingly, the reviewer recommends the manuscript published in Nature Communications after the following issues have been addressed.

Response: We thank the reviewer for recognizing the importance of our work and for the constructive comments.

Comment #2: The authors claimed that additives with multi-functional groups can facilitate the vertical orientation of perovskite and passivate defects, resulting in highly efficient PeLEDs. However, there is no clear evidence to confirm that the vertical orientation of 3D FAPbI₃ perovskite is a key factor in achieving high-efficiency PeLEDs. Therefore, the author should provide more details to explain why the vertical orientation is highly related with device performance.

Response: We thank the reviewer for this comment. The high device performance can be mainly attributed to the low defect density in perovskite film. It has been reported that the sub-grain homogeneity and less grain boundary can reduce the defect density (*Science* 367, 1352–1358 (2020)), so the formation of orientated FAPbI₃ perovskite with high crystallinity can lead to improved device performance. We have added this in the manuscript (Line 195 to 196, Page 10, highlighted).

Comment #3: In Line 64, the authors should attach related reference and add more explanation to demonstrate the relationship between the preferentially perpendicular orientation to substrate and the device performance.

Response: We thank the reviewer for this suggestion. We have added this in the revised manuscript

(Line 65 to 66, Page 4, highlighted).

Comment #4: In Line 171-172, considering the functions of -NH₂ and -COOH groups, the amino acid additives, such as 5-AVA (used in the authors' previous work, *Nature* 562, 249–253 (2018)), will be supposed to be beneficial for obtaining high-efficiency device. Why the authors selected AEAA? Do the additional two oxygen (O) atoms have some extra effect on the crystallization or passivation? The authors should provide more detailed measurement to demonstrate this issue.

Response: Besides the high crystallinity and defect passivation induced by the -NH₂ and -COOH groups, the ether group can form a chelate ring with carboxyl group and unsaturated Pb, which better passivate iodine-vacancy defects. We have added this in the manuscript (Line 199 to 204, Page 10, highlighted).

Comment #5: Xu et al. reported that oxygen (O) atoms in the amino-functionalized additives (EDEA) would reduce the hydrogen-bonding ability (*Nat. Photonics* 13, 418–424 (2019)). The structure of AEAA is quite similar with the EDEA, and the only difference is that the one of the end groups (-NH₂) was replaced by carboxyl (-COOH). Thus, it is important to emphasize the distinctiveness of AEAA.

Response: We thank the reviewer for this comment. We have added this in the manuscript (Line 176 to 177, Page 9 and Line 205 to 207, Page 10, highlighted).

Reviewer #5:

Comment #1: The present manuscript unveils the perovskite crystallization process and the additive role in achieving the perpendicular nanocrystals growth in favor of developing efficient Light emitting diodes (LEDs). Amine, carboxylic acid and amine + carboxylic acid functional group additives influence on 3D perovskite crystallization and passivation were optimized and characterized to form efficient.

The quality of the manuscript is good, despite it possess some research gaps and lack of scientific evidences with respect to author's claims. Major revisions needed along with considerable scientific evidences and discussions.

Response: We thank the reviewer for the positive comments.

Comment #2: Why emission of designed LEDs is limited to 800 nm. Could authors employ this crystallization directing additives in achieving blue LEDs?

Response: We thank the reviewer for this comment. We have tried to fabricate 3D $\text{FAPb}(\text{Br}_x\text{Cl}_{1-x})_3$ blue perovskites. As the mixed-halide blue perovskites suffer from serious phase segregation and poor morphology, it is difficult to evaluate the effect of additives.

Nevertheless, we have employed this approach to red perovskite LEDs. We have introduced AEAA to the $\text{FA}_{0.47}\text{Cs}_{0.53}\text{Pb}(\text{I}_{0.87}\text{Br}_{0.13})_3$ perovskite with emission peak at 693 nm. The XRD data show that the inclusion of AEAA can significantly enhance the crystallinity of red perovskite film (see the below figure). Furthermore, the inclusion of 0.5-ratio AEAA significantly increases the peak EQE of $\text{FA}_{0.47}\text{Cs}_{0.53}\text{Pb}(\text{I}_{0.87}\text{Br}_{0.13})_3$ LEDs from 0.1% to 5.1% and the half-lifetime from 1 to 20 min, without further optimization of the device fabrication process.

Characterizations of red perovskite films and LEDs fabricated with various AEAA amounts. a, XRD data of $\text{FA}_{0.47}\text{Cs}_{0.53}\text{Pb}(\text{I}_{0.87}\text{Br}_{0.13})_3$ perovskite films. **b**, Current density and radiance versus voltage. **c**, Dependence of EQE on current density. **d**, EL spectra. **e**, Stability of LEDs measured at a constant current density of 50 mA cm^{-2} .

Comment #3: The achieved EQE of 19.4% is good, how such achieved EQE can be related with crystallization and passivation alone? Is there any other possible scientific reasons behind the EQE achievement?

Response: Firstly, if we assume the variation of device outcoupling efficiency is negligible, the EQE is mainly related to the PLQE of perovskite films. The inclusion of AEAA additive results in a record

PLQE of ~80% compared with other 3D perovskite films, which is due to the low trap density close to that of perovskite single crystals (see the time-resolved PL measurements in Supplementary Figure 12, *Science* 367, 1352–1358 (2020)). Secondly, the defect density of perovskite film is mainly related to the crystallinity and surface passivation effect (*Science* 367, 1352–1358 (2020); *Nature* 555, 497–501 (2018)). Therefore, we believe that the enhanced EQE of AEAA-based perovskite LED can be attributed to the crystallization and passivation induced by amino, carboxyl, ether groups.

Comment #4: Is vertical growth of perovskite 3D crystals merely related to the H-bond interaction alone? If so, why the authors couldn't achieve the directional growth with other amine additives?

Response: We have shown that the perovskite layers prepared with other amine additives also can induce directional growth of perovskite, exhibiting enhanced crystallinity and regular grains (Supplementary Figure 2).

Supplementary Figure 2. Morphology and crystallinity of FAPbI₃ perovskites with various additives. **a-g**, SEM images of perovskites with amine-group additives. Scale bar: 1 μ m. Butan-1-amine (a), Pentan-1-amine (b), Hexan-1-amine (c), Heptan-1-amine (d), Octan-1-amine (e), Phenylmethanamine (f), 2-phenylethan-1-amine (g). **h**, XRD patterns of perovskites with amine-group additives.

Comment #5: EQE values are still confusing, in abstract it is stated as 19.4% whereas in introduction it is mentioned as 22.2%. Authors should finalize their EQE value and discuss with plausible explanation.

Response: We thank the reviewer for this comment. We have changed the EQE to 22.2% in abstract.

Comment #6: I suggest plotting luminance vs voltage along with radiance, so that it would be better to contrast the LED performance with other literary works.

Response: For the visible LEDs, luminance is useful for human eyes to define the brightness of LEDs. However, the EL peak of our LEDs locates in the NIR regime (~800 nm). We are hard to understand the point of plotting luminance vs voltage.

Comment #7: It is vital to frame the comparison table for the present study with recently published literatures in terms of luminance, EQE, current efficiency, and stability to pronounce the importance.

Response: We thank the reviewer for this suggestion. We have added the comparison of our work with other high-performance NIR perovskite LEDs (Supplementary Table 1).

Supplementary Table 1 Comparison of our perovskite LED with other high-performance NIR devices.

Emission layer	EL peak (nm)	Peak EQE (%)	T_{50} (h)	Current density for stability measurement (mA cm^{-2})	Reference
FAPbI ₃ (AEAA as additive)	800	22.2	19	100	This work
FAPbI ₃ (ODEA as additive)	800	21.6	25	20	3
FAPbI ₃ (5AVA as additive)	803	20.7	20	100	4
NMA-FAPbI ₃ (poly-HEMA as additive)	~800	20.1	46	0.1	5
FAPbI ₃ (pimelic acid as additive)	802	18.6	682	20	6
FA _{0.83} CS _{0.17} PbI ₃ (PPAI as surface passivation molecule)	789	17.5	130	100	7

Comment #8: Various amino group passivating agents were presented in the supporting figure, among which why do authors fix the certain agent on basis of what criteria? From XRD plots, it certainly shows the crystallization differences. Authors should compare the crystallization of the mentioned passivation agents with various doping ratios because one particular doping condition is insufficient in understanding their role in crystallization.

Response: We thank the reviewer for this comment. We agree that other amine-group additives

probably have better effect on crystallization than PAM. Here, we want to show the general effect for amine additives, which can enhance the crystallinity of FAPbI₃, albeit there are minor variations in crystallinity.

Comment #9: Along with the various amino passivating agent doping ratios, crystallite size monitoring with XRD peaks is demanding to reveal the crystallization effects.

Response: By using Scherrer's equation, the crystallite sizes of perovskites with various PAM ratios are calculated (see the below table). It indicates that the PAM can assist the crystallization of much larger crystallites. We have added this in the revised manuscript (Supplementary Figure 1, highlighted).

Crystallite size of perovskites with various PAM ratios.

PAM ratio	0	0.1	0.3	0.5	0.7	0.9
Size (nm)	23	27	28	42	44	60

Comment #10: By the way, the as-compared carboxylic acid agent effects on the morphology and crystalline features also lags with various doping ratios. The data obtained with various passivating agents remains unsatisfactory and it remains complex with no clear outputs.

Response: As responded to the Comment #8, here we want to show the general effect for acid additives, which almost has no impact on the crystallinity of FAPbI₃. Moreover, the investigation of PAC doping ratio has shown that it will not cause significant change in the morphology and crystalline features of perovskite layers (Supplementary Figure 3).

Comment #11: Why authors considering only the end group effects while choosing passivating agents. What will be the effects of the other mid chain groups on perovskite crystallization?

Response: Firstly, the ¹H NMR measurements show that the mid chain group has no interaction with the perovskite precursor. Secondly, we have compared amino/acid additives with different chain lengths and molecular structures, and different end-group additives with the same mid chain (Supplementary Figure 2). It shows that the end group plays the key role on the crystallization process.

Comment #12: The GIWAXS data recorded for PAM and PAC time varies for 7,12s and in particular, why the time variation observed with PAM and PAC agents?

Response: We thank the reviewer for this comment. Due to the strong hydrogen bond between PAM and FA, the deficit of free FA^+ in the precursor solution will suppress the crystallization of perovskite. In contrast, there is weak interaction between PAC and FA, so the PAC-based perovskite forms earlier than the PAM-based sample. We have added this in the revised manuscript (Line 143, Page 7, highlighted).

Comment #13: Why particularly stronger hydrogen bond exists between PAM and FAI? If so other amine additives failed to exhibits such characters, why it occurs predominant in this case?

Response: The strong hydrogen bond is formed through $\text{N-H}\cdots\text{N}$ between PAM and FAI. We have compared different additives, i.e. Butan-1-amine, Octan-1-amine, Phenylmethanamine. The ^1H NMR spectra of FAI solution with different additives show that all the $-\text{NH}_2$ group additives significantly change the resonance signal attributed to the N-H protons of FAI ($\delta = 9.25$ ppm), albeit there are variations of chemical interaction due to the additive choice (Supplementary Fig. 6d). In contrast, the $-\text{COOH}$, $-\text{SH}$, $-\text{OH}$, $-\text{CN}$ group additives result in negligible variations of FAI signals (Supplementary Fig. 6e), suggesting very weak interaction with FAI solution. Therefore, we believe that the chemical interaction between the functional group of additive and FAI plays a critical role in the crystallization of perovskites.

Supplementary Figure 6. d-e, ^1H NMR spectra (DMF- d_7 , 500 MHz) of FAI solutions with amine additives (d) and other functional-group additives (e).

Comment #14: Why titration procedure for FAI equivalent is fixed to 6 and why PbI₂ equivalent is fixed to 2.2 in observing the NMR studies?

Response: We thank the reviewer for this comment. The optimized molar ratio of PAM:FAI:PbI₂ is 0.5:2.4:1. In the titration experiments, the PAM is set as 1, so the FAI equivalent of 6 and PbI₂ equivalent of 2.2 have exceeded their optimized ratios of 4.8 and 2.

Comment #15: How the result do suggests the chemical interaction extent for different amine additives? How to scale the interaction factor by the additives? According to author's discussion, the interaction remains intact, irrespective of chain length aromatic and aliphatic nature of the additives? Kindly recheck the FTIR peaks and discuss with more scientific backgrounds for audience better understanding.

Response: We thank the reviewer for this suggestion.

Firstly, we have shown that the amine additives have stronger hydrogen bond with FAI than other functional-group additives, irrespective of chain length, aromatic and aliphatic nature of the additives (see the response to Comment #13).

Secondly, as FTIR measurement is not surface-sensitive, and it is difficult to determine the weak signal from trace material in the surface of perovskite films, we carried out more surface-sensitive XPS measurements. We find that the signal of Pb 4*f* state in various amine additive samples moves to higher binding energy (see the below figure), which is due to the effect of amine group in determining the crystal growth process, leading to perovskite film with fewer defects.

Pb 4f core-level spectra of perovskite films without, with PAM, Hexan-1-amine (HA) and 2-phenylethan-1-amine (PEA) additives.

Comment #16: Does only the chemical interaction plays role in deciding the out of plane deformation. It is strange that there is no structural and geometric conformational contribution from the added components.

Response: As shown in the response to Comments #4 and 13, the change of size and geometry of additives will cause some variation of perovskite layers, but the chemical interaction between the end group of additive and FAI plays the key role in the crystallization of perovskites.

Comment #17: The scheme didn't work satisfactory in clearing the selective vertical growth of perovskite. It is highly recommended that addition of chemical equations in achieving the molecular ordering stage 1, 2, and 3. How do authors affirm the release of FA^+ in stage 3. The total Figure 4 looks like a mystery as it didn't give more scientific insight with the process involved along with the characterization evidences.

Response: The scheme in Figure 4 is based on the ^1H NMR, ESI-TOF MS and in situ GIWAXS

measurements. The ^1H NMR and ESI-TOF MS measurements indicate the formation of intermediate complex $\{\text{PAM}+\text{FA}+[\text{I}+\text{FA}]_n\}^+$ and $[\text{PAM}+\text{FA}]^+$ in the precursor solution. The GIWAXS measurement during the spin-coating process suggests that due to the deficit of free FA^+ in the PAM precursor solution, the α -phase peak appears later than the PAC sample. Then during annealing process (stage 3), there quickly forms the 3D FAPbI_3 perovskite. This should be due to the release of FA^+ , as the hydrogen bond strengths depend upon temperature.

Comment #18: Authors claims that Carboxylic acid group influence in crystallization is little. The interaction between carboxyl and lead is evidenced with XPS peaks. Why such interaction didn't influence the crystallization, whereas PAM interaction influenced the crystallization significantly. Moreover, I'm curious on the selective passivation on the iodine site instead of other sites present in the perovskite matrices.

Response: We thank the reviewer for this comment.

Firstly, the in situ GIWAXS measurement indicates the crystallinity of the perovskite layers is mainly determined at the very early stage of the spin-coating process (Fig. 2). The ^1H NMR and ESI-TOF MS measurements suggest that the interaction between PAC and perovskite precursor is much weaker than with PAM (Fig. 3 and Supplementary Fig. 6c), so the PAC has little influence in crystallization, but mainly plays the role of defect passivation through the interaction between carboxyl and unsaturation Pb.

Secondly, the ^1H NMR spectra show that the peak pattern of COOH become sharp in the $\text{PAC}\cdot\text{PbI}_2$ solution, suggesting the weak interaction between PbI_2 and PAC (Fig. 3). Moreover, ESI-TOF MS measurement indicates that PbI_3^- exists in the precursor solution instead of Pb^{2+} . Therefore, we believe that PAC should passivate the vacancy of iodine.

Comment #19: The device stability results are compromising with glass-epoxy encapsulation. What would be the stability of LED device without any encapsulation, such figures can add value to the research carried out.

Response: We thank the reviewer for this comment. We have characterized the stability of perovskite LEDs in air without encapsulation (23 °C, ~50% RH). It shows that the AEAA-based device has a T_{50}

of 820 s, which is much longer than the 40 s of device without additive (see the below figure).

Stability of devices measured at a constant current density of 100 mA cm^{-2} .

Comment #20: The light outcoupling efficiency attainment elevates the efficiency. How such outcoupling contributions were studied. The methodology wasn't provided and the characterization regarding those remains vague.

Response: The simulation of outcoupling efficiency was included in the methods (Line 383 to 387, Page 23).

Comment #21: Lack of studies such as PL, lifetime, EL data and how the optimized color purity achieved with respect to crystallization control. Those data along with narrative flow and collective recent references can enhance the quality considerably.

Response: We thank the reviewer for this comment. The PL spectra, PL lifetime of AEEA-based perovskite film and the corresponding EL spectra have been included in the manuscript (Supplementary Fig. 11b, Supplementary Fig. 12d, Supplementary Fig. 14c). The perovskite with AEEA additive shows similar PL/EL spectra as 3D FAPbI₃, but has longer lifetime due to the reduced trap density.

Comment #22: How does the simple traditional 3D perovskite blending methodology worked efficiently, as there are numerous existing methodologies (2D, quasi 2D, nanowires, nanorods and quantum dots) works influential in achieving highly efficient and stable LEDs. How do authors will

employ this strategy in establishing their future works? It is mandatory to describe the bottlenecks associated and how it can be solved in the upcoming works?

Response: We thank the reviewer for this comment. It has been reported that 3D perovskite has the potential to achieve high efficiency by introducing additives to passivate perovskite surface defects and enhance outcoupling efficiency (*Nature* 562, 249-253 (2018); *Nature Photonics* 13, 418-424 (2019)). However, the deep understanding of how additives influence the crystallization process of perovskites is lacking. This work demonstrates a general additive-assisted crystal formation pathway for high-quality 3D perovskites, thus leading to high-performance perovskite LEDs. Moreover, the device efficiency can be further improved by enhancing the PLQE and outcoupling efficiency, which can be achieved by optimizing additives to facilitate the orientated growth, passivate defects and tune the microstructure of perovskite layer. We have added this in the revised manuscript (Line 216 to 218, Page 11, highlighted).

REVIEWER COMMENTS

Reviewer #4 (Remarks to the Author):

The author has addressed all my concerns and the quality of the manuscript has been greatly improved. Hence, I recommend the paper publish in Nature Communications as the current version.

Reviewer #5 (Remarks to the Author):

The present manuscript unveils the perovskite crystallization process and the additive role in achieving the perpendicular nanocrystals growth in favor of developing efficient Light emitting diodes (LEDs). Amine, carboxylic acid and amine + carboxylic acid functional group additives influence on 3D perovskite crystallization and passivation were optimized and characterized to form efficient LED devices.

The quality of the manuscript is improved with reviewer suggestions. Despite the revision, some of the corrections made wasn't satisfactory and herein, I suggest authors to solve the following before publishing this article,

1. The response provided by authors for scheme correction is not convincing. The scheme didn't work satisfactory in clearing the selective vertical growth of perovskite. It is highly recommended that addition of chemical equations in achieving the molecular ordering stage 1, 2, and 3.
2. Why do the stability peak exhibits initial increments and later it degrades, whereas the control sample device degrades gradually.
3. No PL lifetime values were provided. Authors should contrast the reduction in trap density values with lifetime result values.
4. It is well known from the reported works that chain lengths, aromatic/aliphatic nature and mid chain groups have considerable effects in achieving the betterment in the device performance and stability. In addition to it, DFT results compliance proves the involvement of those effects in achieving good results. In this manuscript, the results and discussion part for the stronger bonding with various additives suggest some contradictory results. It is essential to discuss the effect of those contributions when discussing with the different additives. For authors reference, Nature Photonics, 2021, 15, 148-155. Advanced Materials, 2021, 33, 2007855. Science Advances, 2019, 5, eaax4424. Nature Communications, 2021, 12, 644.
5. Authors claimed that the introduction of AEAA to the FA_{0.47}Cs_{0.53}Pb (IO.87Br0.13)₃ perovskite with emission peak at 693 nm. The inclusion of 0.5-ratio AEAA significantly increases the peak EQE of FA_{0.47}Cs_{0.53}Pb (IO.87Br0.13)₃ LEDs from 0.1% to 5.1% and the half-lifetime from 1 to 20 min, without further optimization of the device fabrication process. Why the authors failed to achieve the excellence in EQE of the FA_{0.47}Cs_{0.53}Pb (IO.87Br0.13)₃ LEDs as achieved in FAPbI₃ perovskites LEDs.
6. The point of adding luminance vs voltage curve is to contrast the present study with other reported studies and to understand the turn on voltage characteristics.
7. The crystallite sizes measured using Scherrer's equation of perovskites with various PAM ratios are calculated as 23, 27, 28, 42, 44, 60 nm, respectively. State the Scherrer's equation in the manuscript for the audience clarity.
8. The PLQE values achieved is about 80% and it records high value. I suggest citing some recent references.
9. From the several amine additive studied, authors conclude that induce directional growth of perovskite, exhibiting enhanced crystallinity. The reason for choosing particular PAM additive among other additives still remains vague.
10. To support and strengthen the submitted manuscript, we suggest adding the following recent articles in to your reference list, Nature Photonics, 2018, 12, 681-687. Nature Communications, 2020, 11, 3674. Nature Communications, 2019, 10, 665. Chemical Engineering Journal, 2021, 414, 128866. Nature Communications, 2021, 12, 3472. Chemical Engineering Journal, 2021, 414, 128774.

Point-by-Point Response to Referees

Reviewer #4:

Comment #1: The author has addressed all my concerns and the quality of the manuscript has been greatly improved. Hence, I recommend the paper publish in Nature Communications as the current version.

Response: We thank the reviewer for the positive comment.

Reviewer #5:

Comment #1: The present manuscript unveils the perovskite crystallization process and the additive role in achieving the perpendicular nanocrystals growth in favor of developing efficient Light emitting diodes (LEDs). Amine, carboxylic acid and amine + carboxylic acid functional group additives influence on 3D perovskite crystallization and passivation were optimized and characterized to form efficient LED devices.

The quality of the manuscript is improved with reviewer suggestions. Despite the revision, some of the corrections made wasn't satisfactory and herein, I suggest authors to solve the following before publishing this article.

Response: We thank the reviewer for the positive comments.

Comment #2: The response provided by authors for scheme correction is not convincing. The scheme didn't work satisfactory in clearing the selective vertical growth of perovskite. It is highly recommended that addition of chemical equations in achieving the molecular ordering stage 1, 2, and 3.

Response: We would like to point out that Figure 4 schematically shows the growth pathways of FAPbI₃ perovskite based on the ¹H NMR, ESI-TOF MS and in situ GIWAXS measurements. The ¹H NMR and ESI-TOF MS measurements indicate that there are trace FA⁺, PAM⁺ and [PAM+FA]⁺ in the precursor solution of PAM-based perovskite. The GIWAXS measurement shows that there forms δ -phase FAPbI₃ in Stage I, and low dimensional phase in Stage II. However, the exact component in each stage can not be exactly defined, so the chemical equation can not be added. Nevertheless, we can give a brief description for the crystallization process from stage I to stage III.

In the PAM-based sample, there forms FAPbI₃ (δ phase) in the stage I. In the stage II, the [PAM+FA]⁺ gradually releases the FA⁺, then the δ -phase FAPbI₃ transforms to low dimensional phase ([PAM+FA]₂[FAPbI₃]_{m-1}PbI₄ or [PAM]₂[FAPbI₃]_{m-1}PbI₄). Upon annealing process (stage III), the FA⁺ is fully released to generate the FAPbI₃ (α phase), while the PAM⁺ is attached to the surface of perovskite crystallite to passivate defects.

In the PAC-based sample, there also forms FAPbI₃ (δ phase) in the stage I. However, due to the

relative weak interaction between the PAC and FA, most of FA is released to form the FAPbI₃ (α phase) in the stage II. During annealing process (stage III), the α -phase FAPbI₃ is further produced, and the PAC passivates the vacancy of iodine.

Comment #3: Why do the stability peak exhibits initial increments and later it degrades, whereas the control sample device degrades gradually.

Response: The initial increment of AEAA device is likely due to the trap filling effect of excess mobile ions, and the sequential migration of ions on the perovskite lattice causes the degradation of device during long term operation (*Adv. Mater.* **29**, 1605317 (2017)). For the control device, as the perovskite film has much higher defect densities, the serious ion migration under electrical stress and moisture can degrade the device quickly (*J. Phys. Chem. Lett.* **10**, 6857 (2019)).

Comment #4: No PL lifetime values were provided. Authors should contrast the reduction in trap density values with lifetime result values.

Response: The PL lifetime of perovskite films without or with PAM, PAC and AEAA additives are 0.6, 1.1, 1.0 and 1.1 μ s, respectively, which are consistent with the variation of trap density values. We have added this in the revised manuscript (Supplementary Figure 12, highlighted).

Comment #5: It is well known from the reported works that chain lengths, aromatic/aliphatic nature and mid chain groups have considerable effects in achieving the betterment in the device performance and stability. In addition to it, DFT results compliance proves the involvement of those effects in achieving good results. In this manuscript, the results and discussion part for the stronger bonding with various additives suggest some contradictory results. It is essential to discuss the effect of those contributions when discussing with the different additives. For authors reference, *Nature Photonics*, 2021, 15, 148-155. *Advanced Materials*, 2021, 33, 2007855. *Science Advances*, 2019, 5, eaax4424. *Nature Communications*, 2021, 12, 644.

Response: We agree that the chain lengths, aromatic/aliphatic nature of additives will possibly affect the crystallization process of perovskites. As responded in the previous letter, to verify this effect, we have compared amino/acid additives with different chain lengths and molecular structures, and

different end-group additives with the same mid chain (Supplementary Figure 2). It shows that all the perovskite films with amino-group additives have enhanced crystallinity than those with other functional-group additives, indicating the functional group plays a key role on perovskite crystallization. Therefore, we believe that the choice of amine additives will not significantly influence the oriented crystallization of FAPbI₃ perovskites, as it is mainly determined by the interaction of amino group with FAI in the precursor solution. We have clarified this in the revised manuscript (Line 128 to 129, Page 7, highlighted).

Comment #6: Authors claimed that the introduction of AEAA to the FA_{0.47}Cs_{0.53}Pb(I_{0.87}Br_{0.13})₃ perovskite with emission peak at 693 nm. The inclusion of 0.5-ratio AEAA significantly increases the peak EQE of FA_{0.47}Cs_{0.53}Pb(I_{0.87}Br_{0.13})₃ LEDs from 0.1% to 5.1% and the half-lifetime from 1 to 20 min, without further optimization of the device fabrication process. Why the authors failed to achieve the excellence in EQE of the FA_{0.47}Cs_{0.53}Pb(I_{0.87}Br_{0.13})₃ LEDs as achieved in FAPbI₃ perovskites LEDs.

Response: The red perovskite LED is based on mixed-halide perovskite, which raises the phase segregation issue impeding the fabrication of excellent device. However, the inclusion of AEAA has significantly enhanced the performance of FA_{0.47}Cs_{0.53}Pb(I_{0.87}Br_{0.13})₃ LED, which can indicate the general role of this strategy.

Comment #7: The point of adding luminance vs voltage curve is to contrast the present study with other reported studies and to understand the turn on voltage characteristics.

Response: The luminance is only meaningful for visible light. Specifically, the luminance is calculated from the luminosity function of the human eye, which only responses to wavelength of ~400-700 nm. (https://en.wikipedia.org/wiki/Luminous_efficiency_function). Therefore, it can be used for human eyes to define the brightness of visible LEDs, but can not be adopted in our NIR LED with an EL peak at ~800 nm (which is invisible, literally, no luminance).

Comment #8: The crystallite sizes measured using Scherrer's equation of perovskites with various PAM ratios are calculated as 23, 27, 28, 42, 44, 60 nm, respectively. State the Scherrer's equation in the manuscript for the audience clarity.

Response: We thank the reviewer for this comment. We have added the statement in the revised manuscript (Supplementary Figure 1, highlighted).

Comment #9: The PLQE values achieved is about 80% and it records high value. I suggest citing some recent references.

Response: We have added the references in the revised manuscript (Line 179, Page 9, highlighted).

Comment #10: From the several amine additive studied, authors conclude that induce directional growth of perovskite, exhibiting enhanced crystallinity. The reason for choosing particular PAM additive among other additives still remains vague.

Response: From the characterization of FAPbI₃ perovskites with various amine additives, we find that the end group of additive plays the key role on the crystallization process (Supplementary Figure 2). Thus there is no particular reason to choose PAM among other amine additives.

Comment #11: To support and strengthen the submitted manuscript, we suggest adding the following recent articles in to your reference list, Nature Photonics, 2018, 12, 681-687. Nature Communications, 2020, 11, 3674. Nature Communications, 2019, 10, 665. Chemical Engineering Journal, 2021, 414, 128866. Nature Communications, 2021, 12, 3472. Chemical Engineering Journal, 2021, 414, 128774.

Response: The related references have been added in the revised manuscript.